# Personal, Academic Stressors and Environmental Factors Contributing to Musculoskeletal Pain among Undergraduates Due to Online Learning: A Mixed Method Study with Data Integration

**DOI:** 10.3390/ijerph192114513

**Published:** 2022-11-04

**Authors:** Deepashini Harithasan, Devinder Kaur Ajit Singh, Nur Aqilah Binti Abd Razak, Nadirah Binti Baharom

**Affiliations:** 1Centre for Healthy Ageing and Wellness, Faculty of Health Sciences, Universiti Kebangsaan Malaysia Campus Kuala Lumpur, Kuala Lumpur 50300, Malaysia; 2Department of Physiotherapy, SOCSO Tun Razak Rehabilitation Centre, Melaka 75450, Malaysia; 3WQ Park Health & Rehabilitation Centre, Selangor 47301, Malaysia

**Keywords:** musculoskeletal pain, personal, academic stressor, environmental, factors, online learning

## Abstract

Aim: The COVID-19 pandemic has led to adaptation in teaching and learning methods. There is a possibility that this shift from the classroom to online learning will persist post-pandemic with implications to all involved. We explored the contribution of personal, academic stressors and environmental factors contributing to musculoskeletal pain among undergraduates due to online learning by integrating data from an online survey and one-to-one in-depth interviews. The association between musculoskeletal pain, personal, academic stressors and environmental factors among undergraduates due to online learning was also investigated. Methods: Both quantitative and qualitative methods were used. A questionnaire was completed by 179 undergraduates (34 males and 145 females) aged between 18 to 25 years old. This was followed by an online, in-depth, one-to-one interview among 10 female undergraduates who reported severe musculoskeletal pain. The two sets of findings were integrated using a triangulation protocol. Result: The three most common musculoskeletal pains experienced by undergraduates due to online learning were low back (73.2%), followed by neck (68.7%) and shoulder (58.7%) pain. The six main themes identified from the interviews were: (1) Musculoskeletal pain characteristics; (2) academic issues; (3) difficulties faced by undergraduates due to teaching and learning; (4) emotions towards work/study; (5) work environment; and (6) time spent working at a workstation. Upper back pain was identified to be associated with personal (*p* < 0.05) and most environmental factors (*p* < 0.05). From the triangulation model, it was shown that personal, academic stressors and environmental factors were mainly from the workstation, uncomfortable environment, working posture and time spent at the workstation, which all contributed to musculoskeletal pain. Conclusions: This study showed that exercise, academic stressors, and environmental factors were associated with musculoskeletal pain among undergraduates due to online teaching and learning sessions. There may be a need to integrate an online prevention of musculoskeletal pain education package based on a biopsychosocial model with online teaching and learning for undergraduates.

## 1. Introduction

The novel coronavirus (COVID-19) pandemic changed the daily habits of the population globally [1,2]. The changes are mainly due to lockdown conditions implemented by countries worldwide [2]. In Malaysia, the movement control order (MCO) started on 18 March 2020 [3] and was continued for one year, seven months, and two weeks. The education sector, including higher education, switched from in-person to fully online learning to prevent the widespread transmission of COVID-19 and follow physical distancing measures [1,4]. However, this situation was a completely new challenge to teachers/lecturers and students [2] since they had to adapt and acquire skills to be fully online during teaching and learning activities using either synchronous or asynchronous methods. These methods promote the use of more autonomous methods of learning by the students, without compromising the usual schedule and prior planned workload [1,2]. These abrupt changes may have a negative impact on health status, specifically musculoskeletal pain due to prolonged hours of online learning.

Musculoskeletal pain can be defined as the consequences of repetitive exertion, moderate use of the musculoskeletal system, and work-related musculoskeletal disorders. Musculoskeletal pain may be persistent or develop into a chronic condition, but may also be intermittent with varying intensity at either a single anatomical site or even multiple sites [5]. Consequently, there was a vicious cycle of discomfort, limited daily physical and leisure time activities, and increased stress [2]. Additionally, it may affect students’ academic performance and future working capacity in their transition from university to professional working life [6].

Multiple risk factors may contribute to musculoskeletal pain, such as personal factors, including age and ongoing medical conditions, such as diabetes or rheumatoid arthritis, weight and height, gender and levels of individual physical activeness [7]. Women have consistently been affected by musculoskeletal pain in one or several regions of the body [5]. Additionally, increasing age has been associated with musculoskeletal pain [7]. With age, musculoskeletal tissues show a loss of cartilage resilience, loss of muscular strength, reduced ligament elasticity, increased bone fragility, and fat redistribution leading to a decline in the ability of the tissues to function efficiently [8].

Academic stressor is a known contributing factor of musculoskeletal pain. Some reports pointed out that undergraduates experienced stress and is due to multiple reasons, including a lack of supportive interaction between peers and educators, financial issues [9,10], increased academic workload, and inadequate time to develop knowledge [11]. The online learning process cannot be optimized, especially in courses that require hands-on experience and are designed to include psychomotor learning [1]. Academic stress may interrupt the body’s internal and external environment leading to physiological changes and homeostasis disruption [9]. Moreover, perception of stress and its association with mental health status appeared as significant risk factors contributing to pain in many-body systems [12]. According to James et al. (2018), excellent mental health significantly reduced neck discomfort compared to those who had only a good mental health status. This is further supported by a previous study suggesting that psychological disorders, such as anxiety and depression, can affect the onset of musculoskeletal disorders [13].

Moreover, environmental factors or the non-ergonomic environment, such as prolonged static postures or loading, may cause physical stress, leading to muscle strain, joint imbalance, and soft tissue impairments [9,14]. During online learning, undergraduates spend long hours using laptops, computers and smart devices, which lead to complaints of musculoskeletal pain, namely neck, shoulder and back pain [15]. However, the relationship between these factors and musculoskeletal pain among undergraduates due to online learning is not well established. Comprehensive information pertaining to this matter may provide strategies for preventing musculoskeletal pain due to online learning. Therefore, in the present study, we used a mixed-method quantitative and qualitative design to assess the association between musculoskeletal pain with personal factors, academic stressors, and environmental factors among undergraduates due to online learning.

## 2. Methods

The objectives of this study were to identify the personal factors, academic stressor, and environment factors affecting musculoskeletal pain among university students during online learning. We also examined the association of personal, academic stressor, and environment factors in the onset of musculoskeletal pain among university students during online learning. An online survey was conducted among undergraduates at Universiti Kebangsaan Malaysia, Kuala Lumpur Campus (UKMKL), Kuala Lumpur, Malaysia, followed by in-depth interviews with selected participants. The study was approved by The Research Ethics Committee of Universiti Kebangsaan Malaysia (Ethic no: UKM PPI/111/8/JEP-2021-062). Consent from each participant was taken before data collection.

### 2.1. Study 1: Quantitative Study (Online Survey)

#### 2.1.1. Participants

All undergraduates at UKMKL were invited to participate in this study. For the quantitative study, invitations were made via social media, such as WhatsApp and Instagram. The invitation included a statement regarding the objective of the study and a link to the consent to participate to the questionnaire. The inclusion criteria were undergraduates from UKMKL aged 18 years old and above, and the exclusion criteria were those who did not experience any musculoskeletal pain. This was determined by the respond ‘no’ to all the questions about pain during the previous week in the Standardized Nordic Questionnaire. A total of 179 undergraduates (34 males and 145 females) aged between 18 to 25 years old participated in this online survey. The imbalance between males and females is likely influenced by the gender disparity at the university. However, this sample is representative of the gender distribution of the university. All the questionnaires were in English as the undergraduates are well versed in the language.

#### 2.1.2. Sociodemographic Characteristics

The participants’ sociodemographic characteristics were recorded, including gender, age, race, marital status, education level, average time spent on online learning, the effectiveness of online learning, self-perceived exercise, and time spent per day for exercise with the dominant hand. Musculoskeletal pain was measured using a ‘yes’ or ‘no’ answer, with “yes” indicating that the participants had experienced musculoskeletal pain due to online learning. Physical fitness perception (PFP) is defined as one’s perception of sport competence and physical fitness ability or self-perceived exercise efficacy [16]. Table 1 depicts the frequency and percentage of demographic characteristics, personal factors, academic stressors, and environmental factors of the participants in this study.

#### 2.1.3. Musculoskeletal Pain Assessment

A Standardized Nordic Questionnaire was used to determine musculoskeletal pain. The Nordic questionnaire is a reliable and valid method to screen for musculoskeletal conditions in ergonomic and occupational health contexts [17]. This instrument consists of 40 forced-choice items that help to identify the body areas that cause musculoskeletal problems. The test-retest reliability and validity of this questionnaire ranged from 0% to 23% and 0% to 20% (disagreement), respectively. This is acceptable as a screening tool.

#### 2.1.4. Student Stress Assessment

Student Stress Inventory (SSI) Edition 2019 was used as part of the questionnaire to measure the level of stress among undergraduates. It consists of 40 negative items with 4 subscales; physical (subscale 1), interpersonal relationship (subscale 2), academic (subscale 3), and environmental factor (subscale 4). The assessment was rated via the 2 points Likert scale; mild stress was rated as 1 and moderate stress was rated as 2. The SSI questionnaire had a high content validity with overall scores of 0.805 (80.5%) and high reliability with an overall reliability coefficient of 0.857 [18].

#### 2.1.5. Quality of Working Life Assessment

The Quality of Work Life Surveys 1977–2008 was used to measure the environmental factors. It consists of a list of adverse factors in a work environment used to measure the possible factors that may contribute to musculoskeletal pain. It contains 20 questions with a “Yes” or “No” answer based on the factors given.

### 2.2. Study 2: Qualitative Study (Online In-Depth Interview)

Participants who completed the questionnaire for the cross-sectional study and who were identified with severe musculoskeletal pain were invited to be interviewed. Potential interviewees were contacted by phone to ask for their interest in participating in this study. The online meeting time and platform were arranged once the participants agreed to participate in this study. A total of 10 females (*N* = 10) agreed to be interviewed, all of whom were identified as having pain on multiple body sites.

The researcher facilitated the sessions, guided by semi-structured questions combined with a series of probing questions designed to elicit the factors causing musculoskeletal pain among undergraduates due to online learning, focusing on personal, academic stressors and environmental factors. Questions were also asked based on diagrams adapted from a study by James et al. 2018; (1) *What about the position and height of the monitor and the position of the lower limbs in relation to the height of your desk?* (2) *How about your posture while working?* (Participants were allowed to see Figure 1 and Figure 2 to identify their postures); (3) *On average, how many times and hours do you spend on a computer and doing deskwork per day?* All the questions provide reliable data on workstation and individual characteristics during computer related work [12].

Each interview lasted between 25 to 35 min, was conducted in English, and was recorded using an audio recorder for further transcription and analysis. The questions above were used as the base and additional questions were asked to prompt participants for further answers. No additional interview was carried out when the data reached saturation.

## 3. Data Analysis

The analysis was performed using the Statistical Package of Social Sciences (SPSS) software version 25. For the quantitative study, descriptive analysis was conducted to obtain all variables’ frequencies, mean, SD, and median. This study has three independent variables: Personal, academic stressors status, and environmental factors. Binary logistic regression was conducted to predict the association between predictors (independent variables: Personal, academic stressors status, and environmental factors) and predicted variables (dependent variables: Musculoskeletal pain). Differences with *p* < 0.05 were considered statistically significant.

Meanwhile, for the qualitative study, the audio-recorded interviews were transcribed. The researcher performed thematic analysis on the transcriptions to identify the musculoskeletal pain and its association with personal, academic stressors and environmental factors among the undergraduates due to online learning. Two researchers read through the notes several times to familiarize themselves with the contents. The responses were coded. In this study, transcripts were not returned to the participants for comments. No software was used in this analysis.

The two sets of findings (quantitative and qualitative) were integrated using a triangulation protocol in the final part of the analysis (Figure 3). Themes were compared to establish whether they were in agreement, partial agreement (the two findings complemented one another), dissonant (the findings conflicted), or silent (only one data source contributed).

## 4. Results

### 4.1. Quantitative Analysis

In order to estimate the probability of musculoskeletal pain among undergraduates in UKMKL, a binary logistic regression analysis was conducted. Assumption testing performed before the analysis did not indicate any violations. Table 2 represents the association between musculoskeletal pain with the personal, academic stressors and environment factors. From the analysis, it is shown that neck, elbow and ankle/feet pain are not associated with any of the factors. Upper back pain is identified to be associated with personal (*p* < 0.05) and most of the environmental factors (*p* < 0.05).

### 4.2. Qualitative Analysis

Table 3 shows the demographic information of the participants and the site of pain experienced by them. Each participant was referred to as P1 to P10. Factors related to musculoskeletal pain were categorized into personal, academic stressors and work environmental factors. Six (6) main themes and 28 subthemes have been identified from the interview data. Table 4 shows the number of responses for each subtheme of the main themes.

Based on Table 3, half of the participants did not perform any physical activity, the other five (5) performed physical exercises once or twice a week for about 10 to 30 min per day. From the interview, participants performed exercises, such as simple workouts in their room, skipping or dancing, which did not require them to go out from home. As for BMI, most of them had normal BMI, and only two participants reported being underweight or overweight, respectively.

## 5. Discussion

In this study, we explored the contribution of personal, academic stressors, and environmental factors that were related to musculoskeletal pain among undergraduates due to online learning by integrating data from an online survey and one-to-one, in-depth interviews. While the integration of mixed methods can be challenging, the use of surveys provided a breadth of response, while the interviews provided depth. In addition, musculoskeletal pain and its association with personal, academic stressors and environmental factors were identified. From the triangulation model, it is shown that personal, academic stressors and environmental factors mainly include the type of workstation, uncomfortable environment, working postures and time spent at the workstation, which contributed to musculoskeletal pain.

Low back pain (73%) appeared as the most common musculoskeletal pain, followed by neck (68%) and shoulder (58%) pain in undergraduates in this study. The majority of the undergraduates reported experiencing pain at multiple sites instead of only a single-site. Based on the interview, the time of onset was within the past 3–12 months, which was during the COVID-19 pandemic lockdowns, and students were required to attend their teaching and learning online. During online learning, students spent long hours on their laptops, computers and smart devices leading to several complaints like neck, shoulder and back pain [2,12,19], which is similar to the findings of our current study. It is reported that prolonged sitting, combined with maintenance of static postures aggravated low back pain [15,20]. In addition, prolonged neck flexion with the lack of arm support is linked to neck and shoulder pain [21,22].

In addition, physical activity (PA) also plays a crucial role in the onset of musculoskeletal pain. The current recommendation is an adult should engage in 150 min per week of moderate-intensity aerobic activity or 75 min per week of vigorous aerobic activity, or a combination of both, preferably spread throughout the week. This is equivalent to 30 min a day, five days a week. However, worldwide, one in four adults do not meet this recommendation [23]. Similarly, about two-thirds of the undergraduates in our study did not even meet 30 min of exercise per day. Long hours were spent on online learning, with approximately 35% and 40% reporting being online for five–seven and three–five hours, respectively. It was suggested that low, moderate, and higher exercise intensity are associated with a lower risk of having lower back pain, shoulder and neck pain [24,25].

In Malaysia, the PA level was found to be satisfactory among students based on a study done in 2015 [26], however the reduction in PA currently may be due to the increased sitting time due to the need for online learning, which is linked to greater rates of musculoskeletal pain [2]. Thus, students should be encouraged and educated to engage in PA and exercise during the online learning periods as a strategy to curb musculoskeletal pain.

In regards to the associated factors, LBP was significantly associated with academic stressors and environmental factors (humidity and lack of space). Past studies have shown that mental stress is related to a high prevalence of LBP [27,28], which is consistent with the findings of our current study. According to Shan et al. 2013, the incidence of LBP is lower among students with a higher level of satisfaction with everyday learning [28]. However, in our current study, the sudden transition from face-to-face teaching to online learning, where undergraduates were exposed to new social situations and the maintenance of academic responsibility, in addition to prolonged sitting positions, could have led to LBP and academic stressors. This did not include the extra time used for assignments and other learning task completion. In addition, LBP might be aggravated due to the humidity and lack of space experienced by undergraduates during online learning. Humid conditions had a negative impact on the symptoms experienced by people with chronic pain [29]. In our study, undergraduates sitting at the same spot for long hours without proper ventilation and working environment could have provoked LBP.

Shoulder pain was found to be associated with environmental factors alone (poor or glaring lighting, restlessness in the work environment, time pressure and tight time schedules). Other aspects of the work environment are also important, such as the lighting intensity. The use of computers with prolonged viewing of the screens can lead to eye strain and awkward postures that could eventually lead to neck, shoulder and back pain in the long term [30,31]. These study results support our current study findings. Lighting intensity should, therefore, be adjusted to meet the needed requirements.

Additionally, working in static sitting positions for a prolonged duration with repetitive movements while typing and using a mouse, results in reduced circulation, joint pain and stiffness [28,32], thus feelings of restlessness that are significantly associated with shoulder pain. Excessive screen time due to limited time, date lines and tight schedules may lead to forward head posture, resulting in excessive anterior and posterior curves in the lower cervical and upper thoracic vertebrae, respectively, to maintain balance [33]. Forward head posture may cause an overall imbalance in the musculoskeletal system and excessive loading on the neck, thoracic spine and scapular region [33].

Universities adapted to the new settings as best they could, by trying to maintain the usual schedule and previously established workload of hours with distance, virtual classes, videos, and promoting the use of new methods of learning. However, the challenge went beyond the educators to the students who were required to quickly adapt, acquiring skills in this new situation at their home environment during the unprecedented period. This is portrayed in the triangulation model where personal, academic stressors and environmental factors, mainly from the type of workstation, uncomfortable environment, working posture and time spent at the workstation, had a huge impact on musculoskeletal pain. Implementing PA during lockdowns was difficult as leaving the house or going out to public places was restricted. Additionally, the time required for online study activities and virtual classes made it impossible for undergraduates to learn alternative ways to perform any form of PA at home.

From the triangulation model, undergraduates experienced academic stressors, either mild or moderate stress and this contributed to musculoskeletal pain. One of the reasons was their expectation that the classes were not as they thought they would be and their readiness for the new method was questionable [34,35]. One participant claimed that she felt overwhelmed with the topics, and some claimed that the amount of assignments doubled compared to the face-to-face teaching method. A study by Aherne 2001 suggested that feeling overwhelmed is one of the signs of stress [36]. In addition, the new method of learning impacted the requirement of clinical skills, which is more effective during the face-to-face teaching method [36].

Participants in our study could be stressed as they were unable to go to the labs/clinics/hospitals to practice their hands-on techniques on others and clients. Exposure to stress may trigger biological, emotional and behavioral changes [37]. Among the biological changes, physical symptoms, such as musculoskeletal pain resulting from static postures for prolonged duration and repetitive movements, was demonstrated in students [37]. Addtionally, the extra workload experienced by undergraduates may have led to anxiety, overthinking and forgetfulness. Anxiousness was found to be in the top three highest-ranking signs of stress [10,36,38]. Similarly, the majority of the undergraduate in our study experienced these emotions. In addition, undergraduates complained of a lack of sleep due to the work demands and failure in managing time, such as finishing assignments at a given timeline, recording practical sessions multiple times and studying late nights. Sleep disturbance may result in physical stress on the body, increasing muscle tension, specifically in the central body areas, such as the neck and shoulder, thus increasing the risk of having musculoskeletal pain [39].

Other causes of stress identified were interpersonal relationships. It was found that undergraduates lacked peer interactions. Another problem includes the difficulty of doing group assignments during the pandemic due to difficulty catching up with the other team members through online meetings, as reported in past studies [3,40]. Additionally, in our study findings, it was identified that a lack of support from family members could have contributed to stress. This result is consistent with the findings obtained by the study by Nassr et al. (2020), which reported family to be a distraction when studying at home and is probably due to time management and multitasking, which included house chores [9,41].

Based on the results, it was recognized that the work environment affects musculoskeletal pain among undergraduates during online learning. Kanchanomai et al. (2011) reported that the non-ergonomic positioning of computers may be one of the factors contributing to musculoskeletal pain [42]. Screen positions that are not at the same level as the eye may cause neck flexor fatigue and pain. An excellent ergonomic position is for the computer monitor to be placed directly at eye level or a little lower [12].

Additionally, chair height, where the feet could not reach the floor, was significantly associated with musculoskeletal discomfort. This study is in line with the findings from our study, where the majority of the undergraduates reported placing their feet on the floor and only a small number of them experienced pain in their lower compared to upper extremities [12]. In addition, uncomfortable desks and chairs may contribute to lower back pain due to long hours of sitting [43]. In regards to the home environment, our study results are similar to a previous study by Nassr et al. (2020), where respondents reported that their homes were not suitable for online learning due to noise, no private space, space constrain and sharing computer devices with family members [3].

In this study, we also identified undergraduate coping strategies pertaining to musculoskeletal pain during online learning. A combination of treatment consisting of stretching exercises, rest, massage, ice therapy and medication was generally favorable. These strategies are acceptable as self-management strategies but lack evidence of effectiveness. For example, resting and protecting a painful area when it is painful may be helpful in the short-term, but it is not effective in preventing recurrent and chronic pain [44]. The usefulness of stretches in preventing musculoskeletal pain is still inconclusive, however the effectiveness of physical exercises per se is evidence-based practice [45,46,47]. Medication, including painkillers, heat patches and liniments, were also used by undergraduates in our study. These findings indicate that the coping strategy was directed mainly to the management of pain rather than targeting the contributing factors.

One of our study limitations is that the methodology used limits the generalization of the results where participants were undergraduates from only one university. However, we assume that online learning implementation is somewhat similar in all universities. Additionally, a larger sample size with an even distribution of gender would be recommended in future studies. The present study may serve as a preliminary finding and as a base for a larger study scope.

## 6. Conclusions

In summary, our study findings showed that the most common sites with musculoskeletal pain experienced by undergraduates due to online learning were lower back, neck, and shoulder. Personal factors, academic stressors and environmental factors were associated with musculoskeletal pain that could have affected undergraduates’ efficiency in online learning and reduce their quality of life. Prolonged use of devices and sitting for an extended period also contributes to musculoskeletal pain. In addition, study space, in which environmental factors are uncomfortable and cannot be controlled, can have a negative impact on the health of undergraduates. Several recommendations to be considered with online classes includes the promotion of physical activity and exercises to reduce the risk of musculoskeletal pain. Ensuring a more comfortable learning space with adequate lighting, noise free, and suitable temperature are important to reduce physical stress. Innovative online teaching and learning, such as game-based learning, flipped classroom, problem based learning, case based learning and many other interactive learning methods, could be used by educators to facilitate flexibility, movements and prevent prolonged static postures. In the future, there may be a need to integrate the online prevention of musculoskeletal pain education package based on a biopsychosocial model with online teaching and learning for undergraduates. Finally, this study provides insight into the factors that are associated with musculoskeletal pain during online teaching and learning that could be targeted as preventive measures to reduce the risk of musculoskeletal pain among undergraduates.

## Figures and Tables

**Figure 1 ijerph-19-14513-f001:**
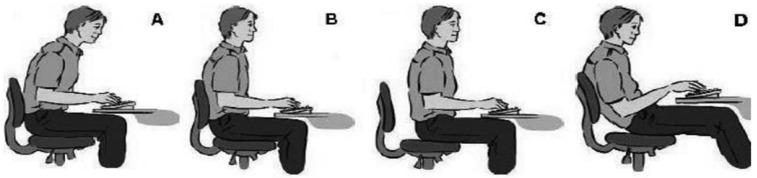
Possible positions in which participants may sit, including variables (altering position at least once per half hour). (**A**). Leaning forward, (**B**). Upright sitting with backrest, (**C**). Upright sitting without backrest, (**D**). Slouching backward. Reprinted with permission from [12].

**Figure 2 ijerph-19-14513-f002:**
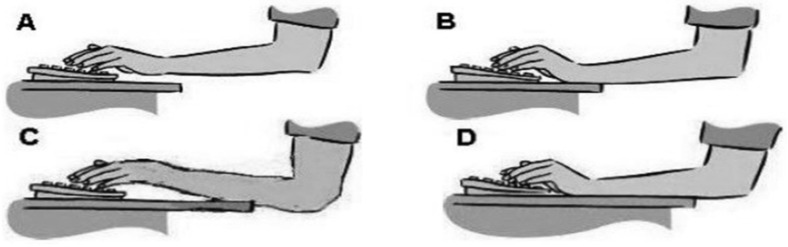
Diagrams selected by participants to describe their wrist and hand positions while typing. (**A**). Wrist and hand floating, (**B**). Keyboard is placed nearer. (**C**). Wrist is extended to reach the keyboard. (**D**). Keyboard is placed further. Reprinted with permission from [12].

**Figure 3 ijerph-19-14513-f003:**
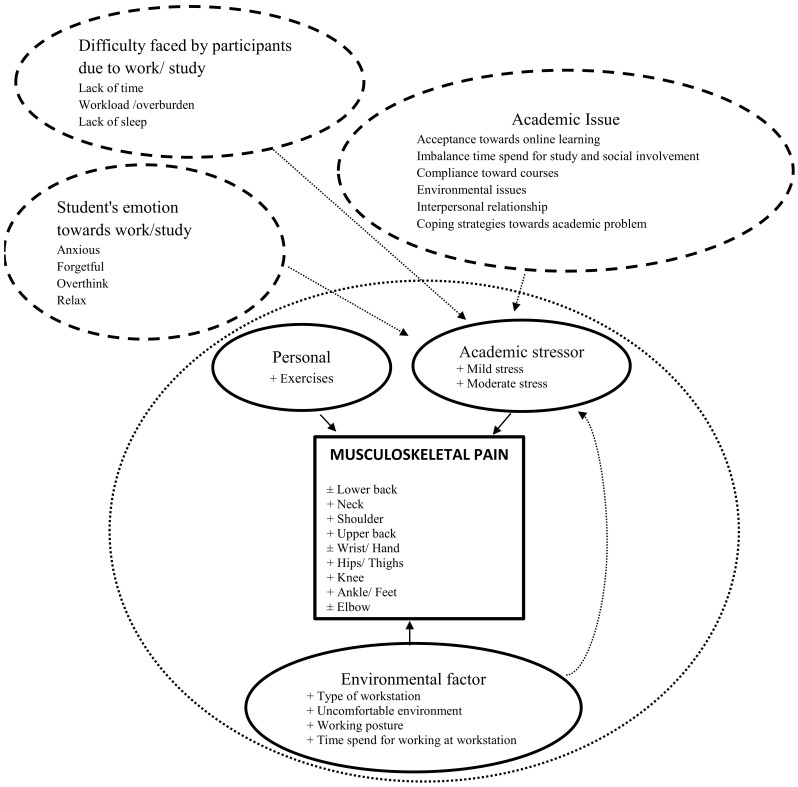
Triangulation model integrating quantitative and qualitative findings (small dots circle), +agreement, ±partial agreement. Supporting data from qualitative findings are included (dashed circle).

**Table 1 ijerph-19-14513-t001:** The frequency and percentage of demographic characteristics, personal factors, academic stressor and environment factors of participants.

Variable		*N* (179)	%
**Age**	18–20 years 21–25 years	29 150	16.2 83.8
**Gender**	Male Female	34 145	19.0 81.0
**Year of study**	1st Year 2nd Year 3rd Year 4th Year	33 29 36 81	18.4 16.2 20.1 45.3
**Program**	Physiotherapy Speech Science Audiology Occupational Therapy Biomedical Science Diagnostic and Radiotherapy Imaging Optometry and Vision Science Nutrition Science Dietetic Environmental Health and Industrial Safety Dentistry Pharmacy	62 15 8 4 11 22 15 4 3 11 4 20	34.6 8.4 4.5 2.2 6.1 12.3 8.4 2.2 1.7 6.1 2.2 11.2
**Marital status**	Single Married	177 2	98.9 1.1
**Dominant hand**	Right-Handed Left-Handed	162 17	90.5 9.5
**Time spent per day for exercise**	10 min 30 min 1 h More than 1 h Not at all	38 66 25 20 30	21.2 36.9 14.0 11.2 16.8
**Average time per day spent on online learning**	1–3 h 3–5 h 5–7 h 7–10 h	29 73 62 15	16.2 40.8 34.6 8.4
**Effectiveness of online learning**	Not all effective Slightly effective Moderately effective Very effective Extremely effective	8 30 122 16 3	4.5 16.8 68.2 8.9 1.7
**Body parts**			
**Neck pain**	Yes No	123 56	68.7 31.3
**Shoulder pain**	Yes No	105 74	58.7 41.3
**Elbow pain**	Yes No	11 168	6.1 93.9
**Wrist or hands**	Yes No	72 107	40.2 59.8
**Upper back pain**	Yes No	89 90	49.7 50.3
**Lower back pain**	Yes No	131 48	73.2 26.8
**Hips and thighs pain**	Yes No	43 136	24.0 76.0
**Knee pain**	Yes No	42 137	23.5 76.5
**Ankles and feet pain**	Yes No	43 136	24.0 76.0
**Personal factors**			
**Self-Perceived Exercise**	Yes No	92 87	51.4 48.6
**Academic stressor**			
**Academic Stressor Status**	Mild stress Moderate stress	87 92	48.6 51.4
**Environment factors**			
**Humidity**	Yes No	81 98	45.3 54.7
**Inadequate Ventilation**	Yes No	57 122	31.8 68.2
**Dirtiness of Work Environment**	Yes No	23 156	12.8 87.2
**Poor or Glaring Lighting**	Yes No	48 131	26.8 73.2
**Restlessness of Work Environment**	Yes No	51 128	28.5 71.5
**Repetitive, Monotonous Movement**	Yes No	72 107	40.2 59.8
**Difficult or Uncomfortable Environment**	Yes No	79 100	44.1 55.9
**Time Pressure and Tight Time Schedule**	Yes No	121 58	67.6 32.4
**Lack of Space**	Yes No	60 119	33.5 66.5

**Table 2 ijerph-19-14513-t002:** Association between musculoskeletal pain (body part) with personal, academic stressors and environment factors.

Variables	Odds Ratio	95% CI	*p*-Value
Shoulder Pain			
**Self-perceived Exercise**	0.693	0.381–1.260	0.229
**Academic Stressor**	0.689	0.379–1.252	0.221
**Humidity**	0.953	0.525–1.733	0.875
**Inadequate Ventilation**	0.696	0.369–1.314	0.264
**Dirtiness of Work Environment**	1.721	0.670–4.418	0.259
**Poor or Glaring Lighting**	3.116	1.464–6.632	0.003 *
**Restlessness of Work Environment**	2.052	1.024–4.114	0.043 *
**Repetitive, Monotonous Movement**	1.439	0.779–2.658	0.245
**Difficult or Uncomfortable Environment**	1.168	0.641–2.130	0.612
**Time Pressure and Tight Time Schedules**	2.576	1.356–4.893	0.004 *
**Lack of Space**	1.491	0.785–2.834	0.222
**Wrist/Hand Pain**			
**Self-perceived Exercise**	0.570	0.312–1.042	0.068
**Academic Stressor**	0.386	0.208–0.717	0.003 *
**Humidity**	1.142	0.627–2.081	0.664
**Inadequate Ventilation**	1.540	0.815–2.911	0.184
**Dirtiness of Work Environment**	0.949	0.387–2.326	0.909
**Poor or Glaring Lighting**	1.943	0.995–3.794	0.052
**Restlessness of Work Environment**	2.606	1.341–5.064	0.005 *
**Repetitive, Monotonous Movement**	2.650	1.427–4.922	0.002 *
**Difficult or Uncomfortable Environment**	2.179	1.185–4.008	0.012 *
**Time Pressure and Tight Time Schedules**	2.262	1.148–4.454	0.018 *
**Lack of Space**	2.034	1.081–3.826	0.028 *
**Upper Back Pain**			
**Self-perceived Exercise**	0.475	0.262–0.862	0.014 *
**Academic Stressor**	0.651	0.361–1.175	0.154
**Humidity**	2.317	1.269–4.228	0.006*
**Inadequate Ventilation**	0.812	0.432–1.525	0.517
**Dirtiness of Work Environment**	2.132	0.855–5.317	0.104
**Poor or Glaring Lighting**	3.463	1.697–7.065	0.001 *
**Restlessness of Work Environment**	2.741	1.389–5.408	0.004 *
**Repetitive, Monotonous Movement**	2.481	1.342–4.584	0.004 *
**Difficult or Uncomfortable Environment**	3.101	1.678–5.732	0.000 *
**Time Pressure and Tight Time Schedules**	1.963	1.036–3.723	0.039 *
**Lack of Space**	2.384	1.258–4.516	0.008 *
**Lower Back Pain**			
**Self-perceived Exercise**	0.963	0.497–1.867	0.911
**Academic Stressor**	0.462	0.234–0.911	0.026 *
**Humidity**	2.906	1.410–5.988	0.004 *
**Inadequate Ventilation**	1.362	0.654–2.833	0.409
**Dirtiness of Work Environment**	1.044	0.386–2.825	0.933
**Poor or Glaring Lighting**	1.553	0.703–3.429	0.276
**Restlessness of Work Environment**	1.479	0.685–3.190	0.319
**Repetitive, Monotonous Movement**	1.930	0.947–3.931	0.070
**Difficult or Uncomfortable Environment**	1.452	0.737–2.860	0.280
**Time Pressure and Tight Time Schedules**	1.976	0.996–3.922	0.051
**Lack of Space**	3.980	1.660–9.539	0.002 *
**Hips and Thigh Pain**			
**Self-perceived Exercise**	0.683	0.343–1.362	0.279
**Academic Stressor**	0.476	0.234–0.970	0.041 *
**Humidity**	1.547	0.777–3.080	0.215
**Inadequate Ventilation**	1.791	0.880–3.645	0.108
**Dirtiness of Work Environment**	2.867	1.155–7.120	0.023 *
**Poor or Glaring Lighting**	1.926	0.924–4.014	0.080
**Restlessness of Work Environment**	1.962	0.951–4.047	0.068
**Repetitive, Monotonous Movement**	2.042	1.020–4.087	0.044 *
**Difficult or Uncomfortable Environment**	1.452	0.730–2.889	0.288
**Time Pressure and Tight Time Schedules**	1.800	0.817–3.966	0.145
**Lack of Space**	1.834	0.907–3.711	0.091
**Knee Pain**			
**Self-perceived Exercise**	0.930	0.466–1.856	0.836
**Academic Stressor**	0.950	0.475–1.897	0.884
**Humidity**	1.871	0.930–3.765	0.079
**Inadequate Ventilation**	1.649	0.804–3.380	0.172
**Dirtiness of Work Environment**	3.696	1.491–9.162	0.005 *
**Poor or Glaring Lighting**	1.122	0.520–2.424	0.769
**Restlessness of Work Environment**	1.005	0.468–2.161	0.990
**Repetitive, Monotonous Movement**	1.309	0.651–2.632	0.449
**Difficult or Uncomfortable Environment**	1.751	0.872–3.514	0.115
**Time Pressure and Tight Time Schedules**	0.946	0.454–1.973	0.883
**Lack of Space**	1.487	0.728–3.036	0.277

Note: Association is significant at the 0.05 level (*).

**Table 3 ijerph-19-14513-t003:** Demographic characteristics of participants.

Participants	Age	Academic Stressor	BMI	Site of Pain	Duration Spends for Physical Exercise
P1	23	Mild stress	Normal	Neck	Never
P2	24	Moderate stress	Normal	Neck, shoulder and upper back	Never
P3	23	Moderate stress	Normal	Neck, shoulder, upper back, low back, wrist, hand, hip, thigh, ankle and foot	Never
P4	23	Mild stress	Normal	Neck, shoulder, wrist and hand	Twice a week (15–30 min/day)
P5	24	Moderate stress	Overweight	Neck, knee and thigh	Never
P6	24	Moderate stress	Normal	Back and hand	Never
P7	22	Moderate stress	Normal	Shoulder, low back and wrist	Twice a week or more (30 min/day)
P8	19	Moderate stress	Underweight	Neck and shoulder	Twice a week or more (1 h/day)
P9	22	Moderate stress	Normal	Wrist and buttock	Once a week (10 min/day)
P10	22	Moderate stress	Normal	Back	Twice a week or more (30 min/day)

**Table 4 ijerph-19-14513-t004:** Themes and subthemes.

Theme	Subtheme	Responses (*N*)	Participants Responds
Musculoskeletal Pain			
(1) Musculoskeletal pain characteristics	(1) Site of pain		**Multiple sites** P3: *Yes I have pain on neck, shoulder, upper back, low back, wrist, hand, hip and thigh, ankle and foot.* **Single site** P10: *I think now more pain at the back.*
	Single site Multiple sites	2 8	
	(2) Time on set		P1: *… experience the musculoskeletal pain since the MCO (lockdowns) last year. When we all started to do online class instead of face to face class I think.* P3: *… It all started about MCO last year because we started online class at that time.* P8: *… it starts during the MCO which is this year. The exact time is during the first time I registered in this university in March.*
	Previous 3–12 months	10	
	(3) Rating scale of pain		P4: *For neck I would rate around 5 and shoulder is about the same as the neck I think. Yes, but the hand is like slightly lower maybe probably 2 to 4 …* P5: *Oh if it is pain the neck I would say 7, if hip and thigh I would give 5 but for the knee is 3 only not more than that.*
	1–3 (low) 4–6 (mild/moderate) 7–10 (severe)	2 8 1	
	(4) Seeking health professional helps regarding pain		**Seeking health professionals** P4: *… I did go to physiotherapy but its like my own choice and the pain is not serious that I need to go to physiotherapy regularly ……* P7: *… for the lower back yes I had go to the clinic because it is very painful …… The pain is on the peak and it’s a prolonged pain.* **Did not seek for health professionals** P3: *I will let the pain resolved by itself because I am a type of person who is rarely went to hospital because of muscle pain. The pain is bearable for me*. P6: *No because it is not too painful*
	Yes No	2 8	
	(5) Causes of pain		**Use of devices and prolonged sitting** P1: *I spend most of the time in front of laptop and just stay in that position in a longer time so it cause the pain* P4: *… cause sometimes I hold the mouse for the whole time and keeping in the same posture so like most probably cause the pain* **Working environment/no proper workstation** P4: *cause currently in my house the chair that I sit is without the back support* P6: *… because when at home, I have many siblings and I do not have proper place to study so we just sit anywhere* **Working posture** P5: *Posture I think also cause the pain, i would say my posture is non-ergonomic …* **Stress** P9: *Firstly, I think because of stress maybe, and then stay in rigid position only …*
	Prolonged sitting Working environment/No proper workstation Use of devices Working posture Stress	7 2 9 5 1	
	(6) Ways to cope pain		**Use medication** P1: *If the pain is bearable I will just put ointment on that area of pain but if the pain is unbearable till I can’t sleep I will put the heat patch to relieve the pain. And so far, I felt relieved after I do that method* **Stretching** P8: *for the neck and shoulder I just do stretching like light stretching only. I felt relieved after a few minutes doing stretching* **Rest** P3: *I take a rest and I’m not putting any hot pack or anything on the site of pain. It works for me the pain reduced by itself* **Combine technique:** **Stretching and rest** P10: *Usually I do stretching or rest. I felt the pain relieved* Stretching and massage P4: *… I would do some stretching or massage and that ways tend to relieve it …* **Stretching and ice** P6: *other than stretching I put ice on my hand to reduce the pain*
	Use medication (ointment, patch, painkiller, ) Stretching Rest	2 2 1	
	Combine techniques:		
	- Stretching and rest - stretching and massage - stretching and ice	3 1 1	
**Academic stressor**			
(2) Academic issue	(7) Acceptance towards online learning		**Agree** P1: *I would say that I am more nervous to present in front of the classroom compared to online using a laptop. If it is about presentation I felt nervous in front of classroom because there will be many people that would see us presenting right? If in front of laptop other people will only hear our voice and none will look at us presenting.* **Disagree** P1: *The thing is for our course we have online exam that require us to communicate with lecturer verbally in direct (viva). In fact, that one is more stressful compared to exam face to face.* P4: *For me I feel that is not really different if presenting in front of camera or in front of the real person but personally if asked for my preference actually I might prefer face to face as long as like we can see each other than should be fine. If face to face I can see most people expression so that I can know like whether they have doubts on my presentation whether they agree on my statement somehow like that.* P8: *I feel nervous during in front of laptop compare to classroom. Because I worried that I might stuck in front of laptop and get more questions from others when they did not understand. At least when I’m in front of classroom I can freely move. But in front of laptop I can’t move I just can stay in one position only.*
	Agree Disagree	6 4	
	(8) Imbalance time spend between study and social involvement	6	P3: *It just the time I spend with family is lesser compare to the time I spend to study and do assignment in the room. I would say that I spend most of the time in the room. I did interact with my friend and do something else other than study but mostly in the bedroom. I spend 1 hour for study usually during the lectures session.* P8: *I think balance for both because usually every day I got myself involve with programme I mean I’m involve with social activity. And I also have time to join UKM program. I would say that I spend about 4 hours for joining social involvement which involve calling my family like that …*
	(9) Compliance toward courses		P1: *I felt stress because when they changed the method of learning into online the assignments become double compare to before. For instance, we supposedly to have clerkship at the hospital but due to online learning to cover for the percentage evaluation, the lecturers will gave us a lot of case study in one time. So, when there is a lot of works and at the same time trying to catch up with the dateline I will become stress* P2: *Yes, I had that feeling before. I felt stress and pressure because I think the course is hard to deal with. I felt stressed because the subject obviously difficult and it takes time to understand what I’ve learnt. In fact, physiotherapy course cannot do online because we more to hands on practice and it’s a bit challenging in this kind of pandemic.* P4: *Kind of when I reach my 3rd year. I think it’s because of the overwhelming topics that you receive at the same time … kind of overwhelming but this situation like continues until last semester so right now is my final semester so it’s okay because now I only have two clerkship subjects and one more somehow like economic lesson. So quite kind of relaxing for me now* P7: *No hahaha so far, I’m okay with it*
	Stress Pressure Overwhelming Positive	7 1 1 2	
	(10) Environmental issue	4	P6: *… Then there is not enough space for me to sit properly and study.* * … noise caming from my other sibling who also study and people who watching TV in the same room … my father loves to clean house so like vacuum … we shared space, my little sister also had online lecturse and she will increase the volume so louder. Even though I’m using earphone but I still heard the noises* *… my neighbour did renovation at their home* P2: *Yes, it is hot it really disturbing, most of the time I think during the noon hour. I felt more energized to do work at the morning compared to the noon.* P8: *sometimes no internet is a bit problem because I’m using available university Wi-Fi here*
	(11) Interpersonal relationship issue	2	P7: *If at home there is a lot of commitment, sometimes I need to do the house chores, my dad are calling for me. Because you know when we refused to do what our parents asked for and they will scold us ha..ha..ha.* P9: *Like there is one time I missed classes or quizzes. … but I think it’s enough if they just do a reminder in group class*
	(12) Coping strategies towards academic problem		P3: *usually, I will start doing my assignment a week earlier than the dateline like I tried to do it a little bit and gradually increase my effort to finish the assignment day by day and I will go through again when it’s the time for submission* P4: … *the most difficult part I will asked my lecturers and also if I was shy to ask some lecturers then maybe I will search it online like certain of the questions or articles and last thing I will asked around my friends then solve the problem by myself* P7: … *the way I cope usually I will watch the recording session again or google or watch youtube and if I still do not understand I will ask my friend.*
	Seek help from lecturers/friends/group discussion Use other sources (eg.: youtube, online, study individually, watching record lecture Prepare earlier than dateline	6 6 2	
(3) Difficulty faced by participants due to work/ study	(13) Lack of time	5	P1: *Yes, because I think in a short period I need to do a lot of work.* P6: *We have to do a lot of thing for just one day like sometimes we feel like neglecting our time to study cause I have to catch the materials for clinic, plan for clinic and then need to rehearse then I feel like there is a lot needed to prepared.*
	(14) Workload /overburden	7	P3: … *especially during the first week, suddenly there is many assignments given by lecturers and I do felt as if I cannot cope with it. How I cope with the problem is I keep myself calm and prioritise which assignments should be done first according to the dateline. If the assignment is urgent need to be submitted early then I will do that first*
	(15) Lack of sleep	9	P2: … *either I’m stayed up during the night or I woke up early in the morning. Usually, I stayed up for 1 to 2 hours. Let say today I’m going to stay up then tomorrow I’m going to wake up early at 4 am in the morning*
(4) Student’s emotion towards work/study	(16) Anxious	7	P2: *Yes, sometimes I am feeling guilty to sleep. If I wanted to take a nap I always think like "do I have enough time to do my work later" or "do I have time to finish my assignment’". After that, I will set an alarm just to wake me up later to do my work.*
	(17) Overthink	1	P3: *Yes but occasionally. It is only happened if there is a lot of assignments need to be done in one particular time. Not only that, if exam is around the corner I will experience that feeling too but this time I will not think about that assignments only but, I will be thinking about the study matters too.*
	(18) Forgetful	1	P4: *Somehow when there is a lot of work often I will remind myself because I’m kind of forgetful, so that’s why I list down all the task and I always scared I missed out any task …*
	(19) Relax	1	P9: *… but these past months I’m not thinking that much …*
**Environmental factors**			
(5) Work environment	(20) Type of workstation		P1: *I always do my work on my study table* P2: *Because I did most of the works on my bed. I am using a portable desk so there is no back support.* P8: *Usually it changes sometimes I do my work on table sometimes on the floor or bed*
	Study table On bed Change workstation: study table, bed and floor	7 1 2	
	(21) Type of device use:		P6: *Usually, I use laptop. But there is one time my laptop problem like last semester so I’m using handphone to type my assignments and I think holding phone for a long period of time cause the pain on my hand …*
	Laptop Handphone and laptop Use 3 of them (handphone, laptop and tablet)	1 8 1	
	(22) Level of eyes and head of screen monitor:		P2: *Level of the screen laptop is slightly lower than my eyes. It is not like I have to slouch but just need to bring my chin down a bit. Because for me if the position is too high it is difficult to type on the keyboard.*
	lower same level	7 3	
	(23) Position lower limbs in relation to desk:		P1: *My lower limb positions are not hanging, it touches the floor* P10: *I sit in crossed leg position or sometimes one leg up hahaha it is very rare for me to put both of my legs on the floor* P2: *Usually I’m sitting in the crossed leg position on the chair …* P9: *My foot is hanging I always sit like uncles at the warung (stall) you know, one leg up position.*
	Foot touch floor Cross-leg sitting One leg up Changing more than one lower limb position	5 1 1 3	
	(24) Working posture		P1: *I always changed my position and I think my position would be between A (leaning forward with kyphotic or flexed thoracic) and D (leaning toward seat, kyphotic and neck protruded).*
	ergonomically recommended position (B) leaning toward seat, kyphotic and neck protruded (D) leaning forward with kyphotic or flexed thoracic (A) upright with no lumbar support (C)	2 2 2 1	
	Changing more than one posture		
	- (A,D) - (A,B,C)	2 1	
	(25) Hand placement on the keyboard		P3: *Usually C because I use laptop stand and my hand position need to be in that way to reach for the keyboard.* P9: *So, my laptop I put on the platform so my hand (show posture-it is like the whole arm are up but not supported just the elbow only)*
	A B C D	1 1 4 1	
	Changing more than one hand placement		
	- (C &D) - (B&A) - others	1 1 1	
	(26) Types of input devices used		P6: *Usually I use the tracking pad. But last time I used to position my hand like (holding a phone e.g.: posture-C hand) and when holding the mouse. So, I feel ache when using for a period of time. For me, using tracking pad is better than using mouse.*
	mouse touching pad both	4 5 1	
(6) Time spend working at work station	(27) Weekdays		P1: *Usually 10 hours per day, during weekday and I spend less time on computer during weekend. Maybe, during weekend I spent only 4–6 h in front of computer per day.*
	4 to 6 hours 7–10 hours	4 6	
	(28) Weekend		
	2–3 hours (short) 4–6 hours (medium) 7–10 hours (long)	3 6 1	

## Data Availability

The data that support the findings of this study are available on request from the corresponding author. The data are not publicly available due to privacy and ethical restrictions.

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
