# Peer review of "Personal, Academic Stressors and Environmental Factors Contributing to Musculoskeletal Pain among Undergraduates Due to Online Learning: A Mixed Method Study with Data Integration"

_ijerph, 2022, doi:10.3390/ijerph192114513_

Round 1
Reviewer 1 Report
Introduction
- It is recommended that the authors undertake a more thorough review of the research that precedes their research interest.
- In this regard, it is recommended that they review studies that have been presented on the development of teaching in times of pandemic and the effects produced on university students.
Methodology
- It is important to clearly state the objectives of the research. Authors are advised to clearly state the general and specific research objective in the methodological section.
- The mixed methodology is appropriate in this work, as well as the triangulation of the results.
- The sample selected is adequate, but not balanced in terms of the gender of the participants (34 men and 145 women). In this sense, it is advisable to comment on the origin of the imbalance between male and female participants in the research.
- We would like to thank the authors for detailing in the methodology section the standardised instruments used for each of the dimensions selected in this study.
- In the semi-structured interview, it is advisable to indicate whether any process of validation by experts has been developed to guarantee the quality of the interview.
- The statistical analysis used is relevant. Not only descriptive, but also binary logistic regression is carried out in order to present the relationship between some selected variables.
- However, as far as the interview analysis is concerned, the dimensions and categories of analysis should be detailed and described in detail.
- In addition, as a recommendation, authors are encouraged to use some software for coding the information. Some of these programmes can be Atlas.ti, MAXQDA, AQUA, etc. These programs offer the opportunity to present the results in a visual way and triangulate between researchers the analysis carried out.
Results
- The frequency and percentage of demographic characteristics, personal and academic factors should be included in the methodology, not in the results section.
- The results are presented separately (descriptive, correlation between variables and categories or topics extracted from the interviews).
- It is recommended that the results section be presented for the purposes of the research. The results sections should be established according to the categories of analysis, triangulating the results obtained in the different instruments.
Conclusions
- It is recommended that the authors present the conclusions according to the objectives established in the research. This would make them clearer.
- At the moment the conclusion is insufficient, it should be better argued.
- In addition, this section should consider the future lines of research or prospective, as well as the limitations detected in the study.
Citations and references and regulations
- Authors should review the presentation of tables and figures in accordance with the guidelines established by the journal.
- Citations are correct
- Most of the bibliographical references used are focused on studies carried out during the pandemic years. They are relevant and linked to the subject matter under study.
- However, it is recommended to include some bibliographical references that could be of interest and that can offer a broader perspective of the work developed in Higher Education.
Ethics of the contribution
The authors have indicated reference to the ethical commitment of this research, bearing in mind that they are working with a sample of students. The authors have provided the acceptance of the ethical code obtained before starting the research they present.
Author Response
Thank you for giving me the opportunity to submit a revised draft of my manuscript titled Personal, academic stressors & environmental factors contributing to musculoskeletal pain among undergraduates due to online learning: A mixed method study with data integration’ to the International Journal of Environmental Research and Public Health (IJERPH). I appreciate the time and effort that you and the reviewers have dedicated to providing your valuable feedback on my manuscript. I am grateful to the reviewers for their insightful comments on my paper. I have been able to incorporate changes to reflect most of the suggestions provided by the reviewers. Attached is my respond to the comments given. Thank you

Reviewer 2 Report
The manuscript entitled "Personal, academic stressors & environmental factors contributing to musculoskeletal pain among undergraduates due to online learning: A mixed method study with data integration" propose an observational study to investigate factors associated with musculoskeletal pain in relation to online learning.
The recent COVID-19 pandemic has shown worldwide the need to propose online courses and teaching/learning modalities to allow students and teachers to continue their activities despite the need to reduce social interaction. In this manuscript, the authors wanted to understand if and how this online modality might affect musculoskeletal pain due to increased screen time.
I have two minor suggestions:
- In the abstract, the authors immediately start with the aim of the study without proposing a background or rationale, that might help the reader to understand the importance of this topic.
- I would expand a little bit the limitations paragraph as, although the study was well performed, due to the multifactorial nature of the outcomes a great sample size should be strongly recommended, and in particular with better sex distribution. As such, based on these findings, the authors can propose that such factors might be associated with msk pain, but it is not possible to draw firm conclusions based on this preliminary data.
Author Response
Thank you for giving me the opportunity to submit a revised draft of my manuscript titled Personal, academic stressors & environmental factors contributing to musculoskeletal pain among undergraduates due to online learning: A mixed method study with data integration’ to the International Journal of Environmental Research and Public Health (IJERPH). I appreciate the time and effort that you and the reviewers have dedicated to providing your valuable feedback on my manuscript. I am grateful to the reviewers for their insightful comments on my paper. I have been able to incorporate changes to reflect most of the suggestions provided by the reviewers. Attached are the responses to the comments given. Thank you.

Reviewer 3 Report
The study is interesting and the topic of focus current. Yet, one problematic aspect of the manuscript is the authors’ attempt to prove causal relationship between online learning and musculoskeletal pain, through a non-experimental design based on participants’ perceptions and self-reported data. Instead, potential relationships among the examined variables can be reported. Besides, additional factors e.g., time spent on digital devices for other reasons (not online learning) are not controlled. Below, I provide some additional comments, which the authors may consider for improving the manuscript.
In the introduction (line 57 onwards), the authors make a reference to multiple risk factors which may contribute to musculoskeletal pain such as personal factors and elaborate on this. It was expected to also find here relevant references on other factors which are central in this study, such as academic stressors and environmental factors. Relevant references are provided in lines 66-76, but it would help if the authors could explicitly mention and identify which are the academic stressors and which are the environmental factors.
In the methods section, it is mentioned that participants were recruited based on given criteria, among those, an exclusion criterion was the absence of an experience of any musculoskeletal pain. It is important to clarify how participants were guided to determine whether they did experience musculoskeletal pain, what does this entail and to which degree. There is a disproportion between women and men in the sample, and only women were interviewed. This potentially affects the results and conclusions drawn.
Later on, in lines 110-111, it is stated that ‘Musculoskeletal pain was measured using a 'yes' or 'no' answer, with "yes" indicating that the participants had experienced musculoskeletal pain due to online learning and vice versa’. This is a bit vague and problematic. How could the participants infer whether there was any causal relationship between musculoskeletal pain and the experience of online learning? The use of the term ‘vice versa’ does not stand here meaningfully. Please, provide reliability coefficients for the Standardized Nordic Questionnaire that was administered. It is not clear how personal and academic stressors status factors were measured in this study. It is assumed that the Quality of Work Life Surveys 1977-2008 was used to measure environmental factors.
Author Response

(The authors gave the same response as above.)

Round 2
Reviewer 3 Report
The authors have addressed to an accepted level most of the comments provided in the first round of the review. Among those, it was about including relevant references on other factors which are central in this study, such as academic stressors and environmental factors, which has been addressed. The authors have also addressed a comment that was provided related to the recruitment of participants and the gender imbalance. Some information has been provided for the reliability and validity of the Standardized Nordic Questionnaire, yet reliability coefficients are not provided. References to the validity of the tool here are not clear, and perhaps not needed since this is a standardized tool. Conclusions have been improved.
Minor comments: Double full stop in the first sentence of the abstract to be deleted. Formatting issues are expected to be double-checked before publication.
Author Response
Thank you for giving me the opportunity to submit a revised draft (round 2) of my manuscript titled Personal, academic stressors & environmental factors contributing to musculoskeletal pain among undergraduates due to online learning: A mixed method study with data integration’ to the International Journal of Environmental Research and Public Health (IJERPH). I appreciate the time and effort that you have dedicated to providing your valuable feedback on my manuscript.
